# The reciprocal relation between morphological awareness and spelling in Chinese: A longitudinal study of primary school students

Liping Li[1]*, Ruiying Li[1], Xinchun Wu[2]

**1** School of Education Science, Collaborative Innovation Center for Fundamental Education Quality Enhancement of Shanxi Province, Experimental Teaching Center of Psychology and Cognitive Behavior, Shanxi Normal University, Linfen, China, **2** Research Center of Children's Reading and Learning, Beijing Key Laboratory of Applied Experimental Psychology, Faculty of Psychology, Beijing Normal University, Beijing, China

* liliping2092@163.com

**Data Availability Statement:** All relevant data are within the manuscript and its Supporting Information files.

**Funding:** Philosophy and Social Science Planing Foundation of Shanxi Province (CN), 2019B204;

## Abstract

Spelling is a literacy skill that must be mastered during children's academic development. It involves a variety of cognitive factors, including morphological awareness. Studies in the alphabet and Chinese systems have shown that there is a close relationship between morphological awareness and spelling. Although there is clearly a significant unidirectional effect of morphological awareness on spelling significantly, few studies have explored the bidirectional relationship between morphological awareness and spelling. This three-time point longitudinal study was designed to investigate the reciprocal effects of morphological awareness and character spelling in Chinese. Participants included 124 children from two primary schools in Mainland China. The students were tracked from first grade to third grade and were administered a battery of tests to measure morphological awareness (e.g., homophone awareness, homograph awareness, and compounding awareness) and spelling to dictation, controlling for IQ, phonological awareness, and orthographic awareness. A structural equation model was utilized to examine the reciprocal relation between the students' morphological awareness and character spelling. Results showed that earlier morphological awareness predicted subsequent spelling abilities from first grade to third grade and spelling in first grade predicted morphological awareness in second grade; however, spelling in second grade did not predict the subsequent morphological awareness in third grade. This study suggests that there is a bidirectional association between morphological awareness and spelling from first grade to second grade in Chinese, and a unidirectional association between morphological awareness and spelling from second grade to third grade. Future studies could examine the causal relationship between morphological awareness and character spelling by using an instructional intervention.

Major Project of the National Social Science Fundation of China, 13&ZD188; Graduate Education Innovation Program of Shanxi in 2020, 2020SY311. The funders had no role in study design, data collection and analysis, decision to publish, or preparation of the manuscript.

**Competing interests:** The authors have declared that no competing interests exist.

## Introduction

Spelling is one of the essential skills of literacy, and is the base and precondition of writing. Although computers and cell phones make typing convenient for us, mastery of spelling is a prerequisite for successful typing. In the processing of spelling acquisition, many cognitive skills are needed, one of which is morphological awareness in the oral language.

The characteristics of Chinese script, which involve one-to-one correspondence between morpheme and character and also include many homophones, homographs, and predominant compounding words [1, 2], inherently reveal a close relationship between morphological awareness and Chinese character spelling. Morpheme is the smallest unit of semantic in language. Morphological awareness refers to the ability to manipulate morphology and morphological structure [3]. Manipulating homophones, homographs, and compounding words in oral language manifests in children's morphological awareness at the level of a single Chinese character and word [1, 4–6]. The Chinese writing system is ideographic and is a relatively semantically transparent language. Characters are the basic units of written Chinese, being composed of strokes according to a set of conventional rules, and Chinese character appears as a square shape [7]; for example, the character of "土" (soil) is composed of the strokes "一", "一", and "一". However, the linkage of sounds and meanings of Chinese characters is largely arbitrary [8]. Although there are many compound characters comprised of a phonetic radical and a semantic radical in Chinese, the information provided by phonetic radicals is unreliable in Chinese character reading and spelling [6]. For example, the compound character "狗" (meaning dog) comprises a phonetic radical "句" and a semantic radical "犭". The pronunciation of the phonetic radical "句" (ju4) different from that of the compound character "狗" (gou3), however, the semantic radical "犭" means animal. Children tend to depend on word meaning in the process of reading and writing. Thus, semantic processing might be closely related to Chinese spelling, as analyzing the morphology and manipulating morphological structure are more important in character spelling.

The dual-route model of spelling [9] proposes that spelling ability is obtained by both a lexical path and a sub-lexical path. "Lexical path" refers to semantic transformation and "sub-lexical path" refers to phonetic-grapheme conversion from the phonetic input to the grapheme output. In alphabetic languages, children can use phoneme-to-grapheme relation in spelling processing by segmenting a word into phonemes and converting the phonemes into the corresponding grapheme. However, Chinese is opaque and has no strict phoneme-to-grapheme correspondence, so children would rely more on meaning than phoneme for spelling. The lexical path could be used in character spelling, and Chinese children may use the semantic system of the lexical path. Mastering and analyzing morphemes can help children distinguish Chinese characters and spell the accurate script meeting the accurate semantic requirements. For example, in the words of "河水"/he2shui3/ (water in river) and "合并"/he2bing4/ (merge), the morphemes "河" and "合" have the same pronunciation of /he2/, however, they have different morphological meanings and graphemes. "河" means river but "合" means together.

This is one potential direction in which morphological awareness might support spelling. The lexical quality hypothesis [10] claims that word representation in mental lexicon depends on the quality of low-level cognitive skills. The higher the quality of morphological representation is, the more accurate the spelling is, and the processing of morphology could affect spelling. Children could rely on morphological awareness in spelling. For both alphabetical languages (e.g., Greek, Dutch, Arabic, and French) and non-alphabetical languages (e.g., Chinese), numerous studies focused on the impact of morphological awareness on spelling. For example, a prior study on Greek [11] suggested that morphological awareness in preschool contributed to spelling in first grade, explaining 8% of the variation of spelling. The study of

Greek [12] showed that morphological awareness contributed uniquely to spelling for children aged 6–9 years in concurrent data, but not to spelling eight months later. Two-way longitudinal across-language research for English, French and Greek in grade 2 [13] showed that morphological awareness as a latent variable uniquely predicted subsequent spelling in the three languages. Dutch research [14] also showed derivational morphology contributed to spelling achievement in grade 6. The results of an Arabic study [15] demonstrated that morphological awareness predicted unique variance in spelling, and root awareness and word-pattern awareness of morphological awareness emerged in Arabic spelling processing. Koh et al. [16] showed that inflectional awareness of French in second grade predicted spelling in third grade. The influence of morphological awareness on spelling may be common across languages in the writing system [7, 11, 17]. Even though Chinese characters are a special ideograms, many empirical studies [1, 4, 6, 11] have found that morphological awareness contributes significantly to reading and spelling.

In Chinese, morphological awareness is a core cognitive construct of literacy for Chinese children [6] and morphological awareness predicted spelling for Chinese dyslexia [18]. Tong and colleagues [19] examined Chinese morphological awareness and spelling for Hong Kong Chinese children and found morphological awareness in the third year of kindergarten explained not only the variance in current spelling but also that of spelling one-year-later. Chinese writing is composed of pictographic and ideographic characters, thus phonetic cues are unreliable, and children are more inclined to rely on grapheme-meaning connections. Therefore, manipulating morphemes is beneficial to spelling the correct graphemes of characters.

At the same time, Kuo and Anderson [20] state that extensive exposure to print could lead to better morphological awareness, suggesting that spelling experiences may contribute to developing morphological awareness. This indicates an alternative possibility that children also develop morphological awareness through their spelling. As outlined earlier, this alternative possibility is pertinent given that there is a gradually increasing number of morphologically complex Chinese characters introduced to students in spelling during primary school education. Spelling morphologically complex characters may aid children in becoming familiar with morphemes and learning about the morphological structure of words. For example, when children spell the characters "扫(sweep) 抓(catch) 拖(drag) 摘(pick)", they will see a common radical "扌" in these characters and infer its meanings to be related with "hand." Spelling this radical leads children to deepen their understanding of the radical morpheme "扌" at the sub-word level.

The possibility that earlier spelling is associated with the improvement of later morphological awareness has received little empirical attention. Other previous work [21] has shown that learning to spell predicted children's performance of morphological awareness subsequently a year later and showed that the experience of spelling did affect their knowledge of morphemes. Nunes et al. [21] argued that the causal relationship between morphological awareness and spelling might be a two-way relationship, whereby literacy skills could promote morphological awareness. Shahar-Yames and Share [22] examined the self-teaching hypothesis of spelling in orthographic learning and showed that the process of recoding a printed word obliges readers to attend to cognitive details and the process of spelling facilitates the establishment of the well-specified representations that are crucial for rapid word recognition. Additionally, the self-teaching hypothesis of spelling [22] predicted that spelling would actually produce superior learning outcomes, such as skillfully manipulating morphology and morphological structure, because of the additional demands placed on the spellers. Therefore, it is necessary to explore the predictive effect of spelling on subsequent morpheme awareness to provide evidence for empirical teaching.

The Theory of Spelling Development [23] suggests that the development of spelling is a series of developmental processes. Brown [23] recognizes that children in the first and second

grades acquire basic spelling skills, such as specific letters representing specific sounds, which may involve phonetic strategies. Subsequently, children in the third and fourth grades are familiar with suffixes, compound words, and homophones. This stage involves more strategies for shape and semantics. Shen and Bear [24] suggest that similar to English children, Chinese children's spelling follows the same development stage. They then found that as the grade increases, children would use morphemes and semantic strategies more frequently. In response to the previous studies, morphological awareness plays an important role in the development of spelling.

Taken together, many studies also have focused on the simple correlation between the two variables or the importance of morphological awareness to spelling. From the perspective of spelling development, although morphological awareness and spelling are closely related, it is unclear whether there is a bidirectional relationship between morphological awareness and spelling over time. And few studies have evaluated the bi-directional nature of this relationship. Understanding the temporal relationships between morphological awareness and spelling is important for obtaining useful information for educational researchers and practitioners. A reciprocal relationship would involve more than correlation at a single time point. The lower grades of primary school represent an important period for Chinese children's mastery of essential characters and spelling rules. Therefore, starting with children in first grade as research subjects, this study used a longitudinal approach to investigate the reciprocal connection between morphological awareness and Chinese character spelling. Taken together, we expected to find a bidirectional temporal relationship between morphological awareness and spelling.

## Materials and methods

### Participants

A total of 149 children (boys 80, girls 69) from two primary schools in the Shanxi province of Mainland China were administered a battery of tasks including morphological awareness (e.g., homophone awareness, homograph awareness, and compounding awareness) and spelling at three time points. Each student's IQ was measured only at the first time point as the control variable. During the three-year study period, a total of 25 participants (17% attrition) were eliminated because of transferring to other schools; thus, complete data of 124 participants were available for later analysis. We compared children who were eliminated and those who remained in the third time point. There were no significant differences by gender [$x^2_{(1)} = 0.48$, $p = 0.49$], age [$t_{(147)} = -0.17$, $p = 0.87$], and IQ [$t_{(147)} = -1.32$, $p = 0.19$]. All children were native Chinese speakers. According to the Chinese teachers' reports, none had any severe developmental language delays.

In Mainland China, children go through three years of kindergarten education before entering primary school. Formal literacy instruction is not given in kindergarten, which begins in the first grade of elementary school. When students enter the first grade, they receive at least two months of pinyin training. Subsequently, teachers instruct students by spelling Chinese characters with pinyin on the blackboard using stroke-by-stroke. At the same time, students are required to imitate spelling.

### Measures

Three morphological skills tasks were administered to test the children's morphological awareness in the current study. Assessments were also administered to determine IQ, spelling ability, phonological awareness, and orthographic awareness.

**Homophone awareness.** The previous test [4] was used to assess children's ability to manipulate the homophone morphemes, including two practices and 12 items. The experimenter orally presented the target morphemes in a compound word, such as the target morpheme /yang2/羊(sheep) in /shan1yang2/山羊(sheep). Children were asked orally to produce one word (e.g., /yang2rou4/羊肉mutton) containing the same meaning as the target morpheme to let children clearly understand what the target morpheme is, but this word (e.g., /yang2rou4/羊肉mutton) is not scored. Thereafter, the children were asked orally to produce another word (e.g., /tai4yang2/太sun) containing a different meaning as the target morpheme and was scored one point. The experimenters encouraged children to produce more homophone words with a different meaning. The more homophone words children produced, the higher the score. There were no maximum scores for this task.

**Homograph awareness.** The previous task [4] was used to test the ability to manipulate and perceive the different meanings of homophones, which was comprised of two practice items and 12 test items. Children were orally presented with a target morpheme in a two-character word and then were asked to name two words using the target morpheme—one word including the same meaning as the target morpheme; another word including the different meaning from the target morpheme. For example, the target morpheme /mian4/面 in /mian4-bao1/面包(bread) was presented to children orally. Two possible correct answers could be /mian4tiao2/面条(noodle) and /mian4rong2/面容(face). The morpheme of /mian4/面 in /mian4tiao2/面条(noodle), which was about flour, had the same meaning as the target morpheme /mian4/面 in /mian4bao1/面包(bread). The morpheme of /mian4/面 in /mian4rong2/面容(face), which was about face, had a different meaning with the target morpheme /mian4/面 in /mian4bao1/面包(bread). /mian4tiao2/面条(noodle) was scored one point. /mian4rong2/面容(face) was also scored one point. The maximum score was 24.

**Compounding awareness.** Using the prior measure [4], a compounding awareness task was administered to assess the skills of extracting and combining morphemes. The experimenter orally presented a scene and asked the children to speak a novel word which included the suitable morpheme and the right structure of morpheme according to the scene. For example, the experimenter said "我们把样子像青蛙的小鸟叫什么?(what should we call a bird like a frog?)" The correct answer was "蛙鸟(frog-bird)". A scale of 0–3 was used to score the children's responses by two raters. When the scores of the same item were inconsistent, the two raters would negotiate with each other to come to an agreement. Eight practice items and 20 test items were included in this task. The maximum score was 60.

**Spelling.** The character dictation task was adopted to test spelling including one practice and 36 test items, which had been used in the previous study [25]. Children were asked to write down the target character on an answer sheet (A4 paper size) after it had been repeated three times. Firstly, the target character in isolation (e.g., 中) was read aloud from a recorder; then a two-character word (e.g., 中国) which included the target word was read; lastly, the target character was read again. All two-character words were selected from standard Chinese textbooks. The difficulty of the words increased gradually. If children could not write down the target character, they were asked to mark a circle on the answer sheet to avoid distractions. Each correct spelling was given a score of one point. The maximum score was 36.

**IQ.** The standardized Chinese version of Raven's Progressive Matrices [26] was used to assess children's non-verbal IQ. Each item comprised a pattern with a missing part and a few options to complete the missing part. Children were required to choose one appropriate option to make the pattern complete. This task had a total of 60 items. The experimenters followed the testing process as outlined in the manual. Each correct answer was given scored as one point. Raw scores were used in the analysis of the IQ assessment results. The maximum score was 60.

**Phonological awareness.** The phoneme deletion task [4] was used to measure phonological awareness, consisting of six practice and 12 test items. The experimenter orally presented a syllable and required children to orally delete the initial, the middle, or the final phoneme from the syllable. For example, "what was left when /u/ was deleted in/chuang2/?" One point was given to each correct response. The maximum score was 12 points.

**Orthographic awareness.** A task composed of four patterns of structuring characters was used to assess orthographic awareness [4]. The assessment included 15 position errors which contained legal components but illegal positions in Chinese; 15 disordered strokes; 15 radical errors which had incorrect components but legal positions; and 45 pseudo characters which consisted of legal components of Chinese in legal positions but were made using artificial characters. All items were randomly arranged on the paper. The children were asked to judge whether each target item was a real character. It was noteworthy that unscored pseudo characters were included as filler material in accordance with orthographic rules. One point was given to each correct response, with a maximum score of 45 points.

## Procedure

This three-year longitudinal study was implemented at the beginning of first grade (Time 1), the beginning of second grade (Time 2), and the beginning of third grade (Time 3) respectively. Spelling and IQ were tested in groups, and three measures of morphological awareness were administered individually in quiet rooms which were provided by the primary schools. Each session of testing was about 30 to 40 minutes long. All measures were administered by trained experimenters who were graduate students of education or psychology majors.

## Ethics statement

The relevant research ethics committee of Beijing Normal University approved the current research. The participant's teachers and parents enthusiastically supported this study. Written informed consent was obtained from the school principals, teachers, and parents of all participants.

## Results

### Descriptive statistics

Missing data and outliers were examined using the regression interpolation method. Descriptive analyses for all the variables were calculated, including means, standard deviations, skewness, kurtosis, rank, and Cronbach's α coefficient as shown in Table 1. The absolute values of skewness and kurtosis were less than one excepting for that of homograph awareness time 1 and compounding awareness time 1. There was no excessive variation of the normality distribution.

To examine the effect of grade level, four MANOVA analyses were conducted. In each analysis, grade was included as a within-subjects factor and independent variable, and homophone awareness, homograph awareness, compounding awareness and spelling were the dependent variables, respectively. Results showed that homophone awareness [$F(2,246) = 166.33$, $p < 0.001$, partial $\eta^2 = 0.58$], homograph awareness [$F(2,246) = 242.73$, $p < 0.001$, partial $\eta^2 = 0.66$], compounding awareness [$F(2,246) = 256.13$, $p < 0.001$, partial $\eta^2 = 0.68$] and spelling [$F(2,246) = 2793.63$, $p < 0.001$, partial $\eta^2 = 0.96$] developed significantly with the development of grade, and the effect of grade was significant.

Table 2 presented a summary of the correlation coefficients among all measures for three time points. Spelling time 1 was not significant closely correlated with homophone awareness, homograph awareness and compounding awareness for three time points, excepting for with

**Table 1. Descriptive statistics for all variables at three time points.**

| | M ±SD | Skewness | Kurtosis | Rank | α |
|---|---|---|---|---|---|
| **1 IQ T1** | 27.63 ± 9.27 | -0.09 | -0.91 | 11–47 | 0.93 |
| **2 Phonological Awareness T1** | 6.11 ± 3.72 | -0.06 | -0.93 | 0–12 | 0.85 |
| **3 Orthographic Awareness T1** | 25.67 ± 9.04 | -0.14 | -0.07 | 0–44 | 0.79 |
| **4 Homophone Awareness T1** | 6.86 ± 3.84 | 1.20 | 1.35 | 0–25 | 0.90 |
| **5 Homophone Awareness T2** | 13.22 ± 4.40 | 0.65 | 0.75 | 4–28 | 0.72 |
| **6 Homophone Awareness T3** | 16.01 ± 6.22 | 0.13 | -0.69 | 3–30 | 0.82 |
| **7 Homograph Awareness T1** | 5.90 ± 3.03 | 0.40 | 0.23 | 0–16 | 0.87 |
| **8 Homograph Awareness T2** | 10.65 ± 2.97 | 0.26 | -0.19 | 3–18 | 0.78 |
| **9 Homograph Awareness T3** | 12.51 ± 3.47 | -0.19 | 0.15 | 2–22 | 0.70 |
| **10 Compounding Awareness T1** | 8.82 ± 8.83 | 1.51 | 1.49 | 0–43 | 0.88 |
| **11 Compounding Awareness T2** | 20.73 ± 10.86 | 0.05 | -0.69 | 0–47 | 0.80 |
| **12 Compounding Awareness T3** | 28.78 ± 10.95 | -0.31 | -0.47 | 1–52 | 0.78 |
| **13 Spelling T1** | 6.05 ± 2.39 | 0.27 | -0.26 | 1–13 | 0.75 |
| **14 Spelling T2** | 16.96 ± 1.75 | 0.44 | -0.07 | 13–21 | 0.74 |
| **15 Spelling T3** | 26.89 ± 3.45 | -0.59 | -0.11 | 18–34 | 0.75 |

Note: IQ: Raven's Standard Progressive Matrices (raw scores); T1: Time 1; T2: Time 2; T3: Time 3.

homograph awareness time 3. No significant correlation revealed that spelling time 1 would not affect morphological awareness. The relations between morphological awareness of three time points and spelling time 2, three were significant, excepting homograph awareness times 1 and 2 and compounding awareness time 1.

 **Cross-lagged analyses.** Controlling for the autoregressive effects of variables, a three-wave cross-lagged model was used to explore the reciprocal relationship between

**Table 2. Correlation coefficients for all variables.**

| | 1 | 2 | 3 | 4 | 5 | 6 | 7 | 8 | 9 | 10 | 11 | 12 | 13 | 14 |
|---|---|---|---|---|---|---|---|---|---|---|---|---|---|---|
| **1 IQ** | - | | | | | | | | | | | | | |
| **2 Phonological Awareness T1** | 0.15 | - | | | | | | | | | | | | |
| **3 Orthographic Awareness T1** | 0.32*** | 0.08 | - | | | | | | | | | | | |
| **4 Homophone Awareness T1** | 0.02 | 0.07 | 0.03 | - | | | | | | | | | | |
| **5 Homophone Awareness T2** | 0.18 | 0.25** | 0.21* | 0.27** | - | | | | | | | | | |
| **6 Homophone Awareness T3** | 0.16 | 0.26** | 0.31** | 0.22* | 0.50*** | - | | | | | | | | |
| **7 Homograph Awareness T1** | 0.25** | 0.25** | 0.16 | 0.24** | 0.33*** | 0.36*** | - | | | | | | | |
| **8 Homograph Awareness T2** | 0.23* | 0.24** | 0.22* | 0.37*** | 0.47*** | 0.40*** | 0.35*** | - | | | | | | |
| **9 Homograph Awareness T3** | 0.19* | 0.18* | 0.08 | 0.34*** | 0.41*** | 0.61*** | 0.40*** | 0.48*** | - | | | | | |
| **10 Compounding Awareness T1** | 0.21* | 0.15 | 0.14 | 0.29** | 0.31*** | 0.25** | 0.47*** | 0.37*** | 0.28** | - | | | | |
| **11 Compounding Awareness T2** | 0.27** | 0.24** | 0.22* | 0.31** | 0.36*** | 0.34*** | 0.37*** | 0.50*** | 0.43*** | 0.59*** | - | | | |
| **12 Compounding Awareness T3** | 0.33*** | 0.22* | 0.21* | 0.31** | 0.35*** | 0.52*** | 0.41*** | 0.43*** | 0.54*** | 0.42*** | 0.61*** | - | | |
| **13 Spelling T1** | 0.07 | 0.19* | 0.01 | 0.02 | 0.03 | 0.13 | 0.09 | 0.18* | 0.20* | 0.09 | 0.06 | 0.09 | - | |
| **14 Spelling T2** | 0.01 | 0.15 | 0.13 | 0.22* | 0.16 | 0.36*** | 0.26** | 0.19* | 0.29** | 0.15 | 0.32*** | 0.22* | 0.29** | - |
| **15 Spelling T3** | 0.16 | 0.43*** | 0.21* | 0.13 | 0.28** | 0.34*** | 0.25** | 0.26** | 0.34*** | 0.14 | 0.30*** | 0.25** | 0.24** | 0.49*** |

Note: *$p < 0.05$

**$p < 0.01$

***$p < 0.001$; IQ: Raven's Standard Progressive Matrices (raw scores); T1:Time 1; T2:Time 2; T3:Time 3.

morphological awareness and character spelling. A structural equation model was utilized for data analyses using the statistical software Mplus 7.11. Morphological awareness was included as a latent variable that was extracted by three measures (homophone awareness, homograph awareness, and compounding awareness) as indicators. Latent variables were used to potentially reduce error. Fig 1 illustrates the model of the reciprocal relationship between morphological awareness and Chinese character spelling controlling for IQ, phonological awareness, and orthographic awareness time 1. Before model modification, the fit indices of the cross-lagged model were $\chi^2$ = 130.73, $df$ = 76, root mean square error of approximation (RMSEA) = 0.07 (90%CI = 0.05–0.10), comparative fit index (CFI) = 0.91, Tucker-Lewis index (TLI) = 0.85, standardized root mean square residual (SRMR) = 0.05. According to the recommendations of good fit indexes [27], the ratio of $\chi^2$ to $df$ is smaller than 2, CFI and TLI values larger than 0.90, RMSEA and SRMR values smaller than 0.06. Therefore, in this model, RMSEA was higher and the fit index of TLI was lower. According to the modified index of the model, when compounding awareness time 1 was allowed to correlate with compounding awareness time 2, and compounding awareness time 2 was allowed to correlate with compounding awareness time 3 in the modified model, the fit indices of the modified model were $\chi^2$ = 103.23, $df$ = 74, RMSEA = 0.06 (90%CI = 0.03–0.08), CFI = 0.94, TLI = 0.92, SRMR = 0.05, and the fit indices were better. The modified model was ultimately adopted. The results of the structural equation model showed that there was a bidirectional relationship between morphological awareness and spelling from grade 1 to 2. There was also a unidirectional relationship between morphological awareness and spelling from grade 2 to 3 such that earlier morphological awareness contributed to the subsequent spelling, but earlier spelling did not contribute to the later morphological awareness.

## Discussion

### Effect of morphological awareness on spelling

The current three-time point longitudinal study was consistent with the prior Chinese study [18] and alphabetic language studies [11, 28], demonstrating that morphological awareness was closely related to spelling and had an important role in learning to spell. This result

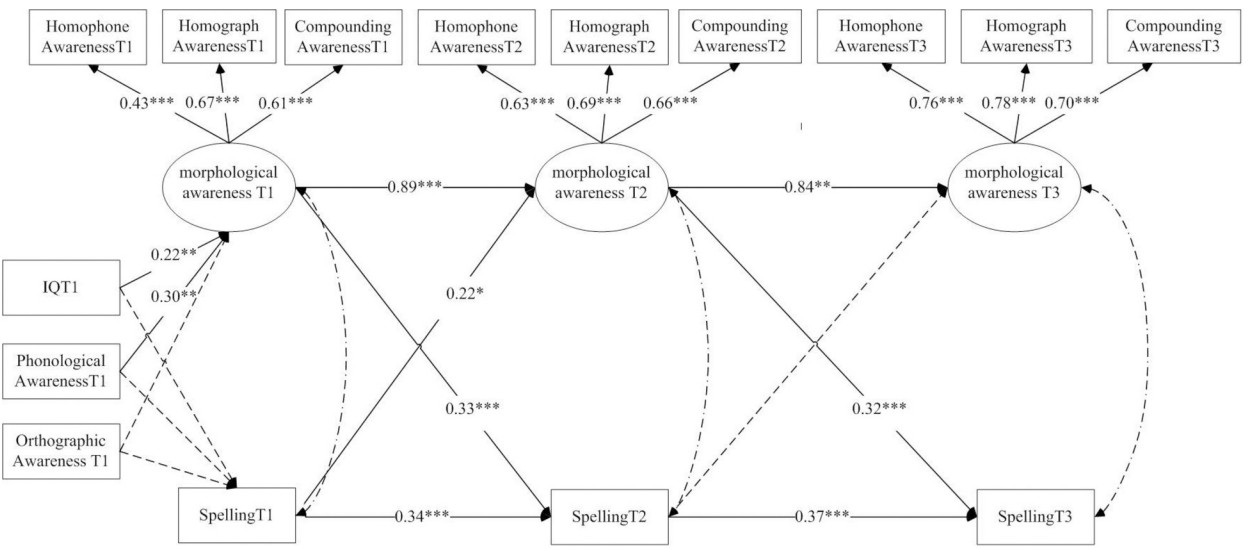

**Fig 1. Model of the reciprocal relationship between morphological awareness and spelling.**

showed that the role of morphological awareness on spelling is universal, applying to both alphabetical languages [15, 29] and ideographic Chinese [18]. The current findings add to the growing body of evidence that cognitive-language skills (e.g., morphological awareness) contribute significantly to subsequent spelling for lower grades. This result indicates that there is a lexical path or semantic transformation in the processing of character spelling [9], and children extract the correct grapheme according to the meaning of the character in the word. Children could determine the grapheme of a target character by distinguishing the morpheme and meaning corresponding to the target syllables in two-character words.

The current study showed that proficient morphological awareness in the early period would improve later spelling ability. On the one hand, spelling is an elaborate process of outputting. According to the lexical quality hypothesis [10], high-quality representations of low-level cognitive skills could benefit from word representation. Higher quality morphological representations help children more skillfully distinguish and operate morphemes; thus children would expertly use this ability in subsequent grapheme outputting of spelling. On the other hand, the characteristics of Chinese determine the importance of morphemes in the processing of character graphemes. Chinese is a logographic writing system, and semantic clues of words are more reliable than phonetic cues [6]. Children prefer to use morphology to output the graphemes in the processing of words. In addition, more homophones in Chinese make children use the morphology and semantic clues to discriminate the form of Chinese characters when children spell these homophones. To sum up, prior morphological awareness contributed to subsequent Chinese character spelling.

## Effect of spelling on morphological awareness

This study suggested that spelling in first grade did significantly predict morphological awareness one year later (in second grade) after controlling for IQ, phonological awareness and orthographic awareness. The results from first grade to second grade did converge with prior intervention research of alphabetic languages [21] and the self-teaching hypothesis of spelling [22]. As demonstrated in the findings of Nunes et al. [21], the better children are able to spell characters, the more aware they become of morphology. One explanation would be that awareness of language (e.g., morphological awareness) is likely to be an abstract form acquired in language learning (e.g., spelling). Children may learn statistically through their spelling experience to obtain abstract morphological awareness, and self-teaching of spelling [22] may occur. From first grade to second grade, Chinese children were required to master more than 1000 characters, and word recognition and spelling are major tasks for these students. First graders must master many words and have extensive exposure to print, and advanced spelling could lead to better underlying morphological awareness. According to the Theory of Spelling Development [23], it is a critical period of spelling development from first grade to second grade. Children must master a large number of Chinese characters and have had extensive exposure to print at this stage. Thereafter, frequent spelling experience and some formal guidance can enable children to acquire abstract awareness, i.e., morphological awareness. Morphology and semantic strategies become increasingly important to children as they grow older. Thus, spelling skills facilitate the subsequent development of morphological awareness.

Additionally, the current study showed that spelling in second grade did not significantly predict morphological awareness one year later (in third grade) after controlling for IQ, phonological awareness, and orthographic awareness. One possible reason is the change of the spelling development stage. The Theory of Spelling Development indicates that children already have some spelling experience from the first stage to the second stage. Chinese children have mastered about 1,300 Chinese characters. Spelling experience could help children perceive

orthographical rules. During the second spelling stage, children have developed a higher level of orthography, which is close to the adult level in Chinese [30]. Children who are more familiar with spelling rules and spelling action achieve to the level of automation. The individual differences in spelling become smaller, leading to the explanation of less variance of morphological awareness. Therefore, spelling may not contribute to the morphological awareness.

Another possible reason is the difference of spelling development rate. The spelling development rate is fast in the first grade, but it slows down in the second grade because children are required to manipulate the higher level cognitive tasks, such as writing structure and thinking [31]. The development of morphological awareness would need a broader context of reading-related skills. A slower spelling development does not provide a broader knowledge background. Children may gain more cognitive resources and linguistic skill by reading than spelling in second stage. Therefore, reading may play a more important role in the development of morphological awareness. Previous study [25] has also found that reading affects the development of children's morphological awareness. The development of morphological awareness could be more affected by reading than by spelling. Hence, the subsequent morphological awareness was not significantly affected by the slower spelling development. The current study found that there is the unidirectional relationship between morphological awareness and spelling from the second to third grade, and the unidirectional relationship might be unique in Chinese. This conclusion also needs to be confirmed with further studies in other languages.

## Implications and future research

Grasping words and scripts is the basis of reading and writing. Understanding the reciprocal relation of morphological awareness and spelling in Chinese could verify the universality of spelling processing theories and promote strategies for spelling in instruction practice. The results of this study are of great significance to the pedagogy. Improving the ability to master spelling is one of the essential tasks of literacy instruction in primary school. The ability to spell could be enhanced by promoting morphological awareness to deeply remember words and seeking ways to improve methods of rote memorization of spelling.

There are two limitations of this study. On the one hand, only the performances of young students in low grades of primary school were investigated in the current research, and the relationship of morphological awareness and spelling in higher graders is not clear. With the development of literacy, the relation of morphological and spelling may change. So, future studies would investigate students in higher grades to describe the role of the relation of morphological awareness and spelling in the whole educational stage of primary school. On the other hand, this study only investigates the correct spelling of Chinese characters, and the errors types were not analyzed. Investigating spelling error types foster a foundation for the improvement of spelling ability and the development of orthographic awareness. Therefore, it should be considered as the meaningful and interesting field that can be discussed in a future study.

## Supporting information

**S1 Data.**
(SAV)

## Acknowledgments

We sincerely thank lovely children, teachers and their parents for their participation. We thank Yahua Chen, Qiong Dong, Thi Phuong Nguyen, Minglu Zheng, Ying Zhao, Peng Sun, Yuan Ding, and the graduate students of Shanxi Normal University for their help with data collection.

## Author Contributions

**Data curation:** Liping Li, Ruiying Li, Xinchun Wu.

**Project administration:** Xinchun Wu.

**Writing – original draft:** Liping Li.

**Writing – review & editing:** Liping Li, Ruiying Li.

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
