## [Decision Letter · Decision Letter 0]

24 Apr 2020

PONE-D-20-01004

The reciprocal relation between morphological awareness and spelling in Chinese: a longitudinal study of primary school students

PLOS ONE

Dear Mrs Li,

Thank you for submitting your manuscript to PLOS ONE. After careful consideration, we feel that it has merit but does not fully meet PLOS ONE’s publication criteria as it currently stands. Therefore, we invite you to submit a revised version of the manuscript that addresses the points raised during the review process.

We have received reviews from two experts, who are in agreement that this is a technically sound study that can make a valuable contribution to the field. Reviewer 1 made some straight-forward suggestions for clarifying the manuscript and enhancing its impact. Reviewer 2 raised some more substantive uncertainties about how the tasks were administered and scored. Reviewer 2 also recommended collaborating with someone to improve the clarity of the phrasing in the manuscript. In addition, I ask that you verify that your manuscript is consistent with the STROBE guidelines for reporting (checklist: https://www.strobe-statement.org/fileadmin/Strobe/uploads/checklists/STROBE_checklist_v4_cohort.pdf).

We would appreciate receiving your revised manuscript by Jun 08 2020 11:59PM. To enhance the reproducibility of your results, we recommend that if applicable you deposit your laboratory protocols in protocols.io, where a protocol can be assigned its own identifier (DOI) such that it can be cited independently in the future. For instructions see: http://journals.plos.org/plosone/s/submission-guidelines#loc-laboratory-protocols

We look forward to receiving your revised manuscript.

Kind regards,

Daniel Mirman

Academic Editor

PLOS ONE

Journal Requirements:

1. Thank you for inlcuding your funding statement;"The funders had no role in study design, data collection and analysis, decision to publish, or preparation of the manuscript."

4. Please remove your figures from within your manuscript file, leaving only the individual TIFF/EPS image files, uploaded separately.  These will be automatically included in the reviewers’ PDF.

Reviewers' comments:

Reviewer's Responses to Questions

**Comments to the Author**

1. Is the manuscript technically sound, and do the data support the conclusions?

Reviewer #1: Partly

Reviewer #2: Yes

2. Has the statistical analysis been performed appropriately and rigorously? 

Reviewer #1: Yes

Reviewer #2: Yes

3. Have the authors made all data underlying the findings in their manuscript fully available?

Reviewer #1: No

Reviewer #2: No

4. Is the manuscript presented in an intelligible fashion and written in standard English?

Reviewer #1: Yes

Reviewer #2: No

5. Review Comments to the Author

Reviewer #1: This is a very strong manuscript that shows potential to be of great interest to a broad audience of researchers - both in the fields of reading and spelling development and those interested in the applicability of models that had been predominantly developed in alphabetic languages (such as English) to other languages (reading and spelling research has been dominated by models based on English speakers, yet it is important to disentangle linguistic influences from cognitive processes). This is a well-designed study, and the findings will move the field further. However, there are a couple of recommendations that may enhance the quality of this manuscript:

* The relevance of this study could be augmented by linking it to universals in reading development. Chen & Pasquarella have a chapter about reading acquisition in Chinese in Verhoeven and Perfetti's (2017) book, Learning to read across languages and writing systems. This chapter could be helpful in situating this study in the broader field, so its impact can be enhanced.

* It would be helpful to provide motivation for the longitudinal aspect of the study (the design itself is a strength). This may be accomplished by discussing models of spelling development. That may also help with the finding that morphological awareness and spelling share a bidirectional relationship in grades 1 and 2, but unidirectional to grade 3.

* In the method section, it would be helpful to have a little more information about the instructional context. For example, how many years of instruction preceded first grade? What was reading instruction like (e.g., was pinyin included? was a look-say method used?).

* It would be helpful to have a little more information about the measures. What were the maximum scores? Were they used and validated in past research? The description of the measures themselves were also a little challenging to follow.

* The tables would be easier to follow with the variable names written out in full.

* The interpretation of the findings would be strengthened if they were connected to broader models of spelling development and universals. This is a strong study, in and of itself. However, it would benefit from being connected to broader models of learning to spell and how the linguistic features of Chinese may lead to different cognitive patterns of performance.

Reviewer #2: I believe that this study could makes a meaningful contribution to the literature on the relationship between spelling and morphological awareness in Chinese. However, some revision is necessary before it is ready for publication. There are some parts that I do not understand well enough to be able to meaningfully evaluate whether the conclusions drawn are supported by the data. The manuscript should be revised in collaboration with a native speaker of English familiar with the topic of the paper. There are some cases where it is difficult for me to tell whether the problem is one of theoretical imprecision or simply a matter of wording.

My primary concerns about this paper have to do with the measures.

The homophone task (page 9) is described in two ways that seem to contradict each other: In lines 175-177 the description is as follows: “This test would assess children’s ability to manipulate the homophone morphemes which owned the same pronunciation and the different meaning and grapheme,” This makes it sound like the child is being asked to provide words with containing a different but homophonous character. The example given later in the paragraph is consistent with this description. However, the following lines (177-179) say: “The experimenter orally presented the target morphemes in a compound word, then asked children to produce other words using the target morpheme.” The phrase “the target morpheme” would seem to seem to imply that the child is supposed to produce words with the same morpheme/character. This may be an imprecise use of the phrase “target morpheme” - but such an error should not occur in a paper on morphology. Furthermore, it is not specified how a response would be scored if the child produced a word using the same character and meaning, or a word using the same character with a different meaning (in this case, “ocean'). If a correct response requires the child to produce a word with a different character – i.e., a different spelling – then this test, though administered orally, is actually to a substantial degree a spelling test.

Likewise, from the instructions for the homograph test, it appears that this test requires the child to produce a word that is written with the same character, but a distinctly different meaning. It is not specified how a response would be scored if the child produced a word with a different, homophonous character. If in fact to be correct the child must produce a word which contains the same character, but with a different meaning, this too is at least in part a spelling test.

About the spelling test (pages 10-11): I can imagine different types of spelling errors in Chinese, that would relate to morphological awareness in different ways. For example, a child might spell a word with the wrong but homophonous character (one with the same sound, but a different meaning). And in this case, there is a further distinction between homophonous characters sharing the same phonetic component, and homophonous characters with nothing in common. Another type of error might be giving a character graphically similar to the correct one. A third type of error might be missing or misplacing a stroke in a character. Which types of errors actually occurred in the data? Might a more nuanced scoring system, distinguishing among error types, lead to more insight?

I did not clearly understand the explanation of why there was not an effect of spelling on later morphological awareness after grade 1.

6. PLOS authors have the option to publish the peer review history of their article (what does this mean?). If published, this will include your full peer review and any attached files.

Reviewer #1: No

Reviewer #2: No

---

## [Author Response · Author response to Decision Letter 0]

23 Aug 2020

We would like to express our great appreciation to reviewers and editor. Those comments are very helpful to our manuscript.

---

## [Editor Report · Decision Letter 1]

16 Nov 2020

The reciprocal relation between morphological awareness and spelling in Chinese: a longitudinal study of primary school students

PONE-D-20-01004R1

Dear Dr. Li,

We’re pleased to inform you that your manuscript has been judged scientifically suitable for publication and will be formally accepted for publication once it meets all outstanding technical requirements. I apologise for the delay. Unfortunately, the reviewers of your initial submission were not available to review your revised manuscript and after extensive effort, I was not able to secure new reviewers. Therefore, I have made this decision based on my careful reading of your manuscript and your response to reviewers.

Kind regards,

Daniel Mirman

Academic Editor

PLOS ONE
---

## [Editor Report · Acceptance letter]

9 Dec 2020

PONE-D-20-01004R1 

The reciprocal relation between morphological awareness and spelling in Chinese: a longitudinal study of primary school students 

Dear Dr. Li:

I'm pleased to inform you that your manuscript has been deemed suitable for publication in PLOS ONE. Congratulations! Your manuscript is now with our production department. 

Kind regards, 

on behalf of

Dr. Daniel Mirman 

Academic Editor

PLOS ONE